# *Cladosporium*—Insect Relationships

**DOI:** 10.3390/jof10010078

**Published:** 2024-01-19

**Authors:** Rosario Nicoletti, Elia Russo, Andrea Becchimanzi

**Affiliations:** 1Council for Agricultural Research and Economics, Research Center for Olive, Fruit and Citrus Crops, 81100 Caserta, Italy; 2Department of Agricultural Sciences, University of Naples Federico II, 80055 Portici, Italy; elia.russo@unina.it (E.R.); andrea.becchimanzi@unina.it (A.B.); 3BAT Center—Interuniversity Center for Studies on Bioinspired Agro-Environmental Technology, University of Naples Federico II, 80055 Portici, Italy

**Keywords:** Cladosporiaceae, endophytes, entomopathogens, mycophagy, symbiosis, species complexes

## Abstract

The range of interactions between *Cladosporium*, a ubiquitous fungal genus, and insects, a class including about 60% of the animal species, is extremely diverse. The broad case history of antagonism and mutualism connecting *Cladosporium* and insects is reviewed in this paper based on the examination of the available literature. Certain strains establish direct interactions with pests or beneficial insects or indirectly influence them through their endophytic development in plants. Entomopathogenicity is often connected to the production of toxic secondary metabolites, although there is a case where these compounds have been reported to favor pollinator attraction, suggesting an important role in angiosperm reproduction. Other relationships include mycophagy, which, on the other hand, may reflect an ecological advantage for these extremely adaptable fungi using insects as carriers for spreading in the environment. Several *Cladosporium* species colonize insect structures, such as galleries of ambrosia beetles, leaf rolls of attelabid weevils and galls formed by cecidomyid midges, playing a still uncertain symbiotic role. Finally, the occurrence of *Cladosporium* in the gut of several insect species has intriguing implications for pest management, also considering that some strains have proven to be able to degrade insecticides. These interactions especially deserve further investigation to understand the impact of these fungi on pest control measures and strategies to preserve beneficial insects.

## 1. Introduction

Fungi in the genus *Cladosporium* (Dothideomycetes, Cladosporiaceae) are ubiquitous and reported from any terrestrial and marine substrate, including all kinds of living organisms [1]. This is linked to their profuse sporulation, which allows the spread of conidia through atmospheric agents over long distances. Thus, the mere isolation of these fungi from plants and animals does not necessarily imply a symbiotic association. However, generality is not a rule, and in many cases, this pervasiveness subtends either occasional or more systematic biotic relationships that influence the fitness of the associated organisms in diverse ways [2,3,4].

The range of interactions is particularly broad in the case of insects, with a various case history of antagonism and mutualism described so far. The available information on the relationships between *Cladosporium* and insects is examined in this paper, with the aim of shedding light on their biological/ecological assumptions, as well as on the ways and circumstances by which they may affect pest control strategies.

## 2. Taxonomic Aspects and Occurrence

Classification of fungi of the genus *Cladosporium* is problematic because of the infrequency of the teleomorphic stage and the absence of outstanding morphological differences in the conidial structures. Indeed, culturing and microscope observations only allowed a poorly rigorous separation of taxa, some of which are assumed to represent collective species, or ‘species complexes’ (s.c.); namely, *C. cladosporioides*, *C. herbarum* and *C. sphaerospermum*. The introduction of biomolecular tools in fungal taxonomy has enabled mycologists to resolve these aggregates with the separation of new entities so that as many as 169 species were recognized in the fundamental revision by Bensch et al. [1]. However, this number is continuously increasing following the finding and classification of novel isolates from all sorts of substrates and environmental contexts.

In this paper, we consider the findings of *Cladosporium* according to the updated nomenclature as far as possible. In fact, studies reporting species identification based on the sequencing of DNA markers are still a minority, and most of them only consider rDNA-ITS sequences, which have proved to be insufficient for a correct classification [1]. In this respect, the approximate phylogenetic reconstruction provided in some reports (e.g., reference [5]) significantly illustrates how identifications based on ITS only are merely tentative.

Data on the occurrence of *Cladosporium* species in association with insects resulting from the examination of the available literature are summarized in Table 1. In total, 303 entries, corresponding to the finding of these fungi on 171 insect species belonging to 16 orders, are included in this long list. On the fungus side, the species was not identified for about 46% of the entries (*Cladosporium* sp.), confirming classification to be quite a problematic aspect within this genus. When identification was accomplished, a total of 32 *Cladosporium* species were recognized. Among them, the three classic species mentioned above stand out in numerical terms; particularly, there are 56 reports for *C. cladosporioides* from 55 insect taxa belonging to 11 orders. This is not surprising considering that this is a polymorphic s.c. in which the existence of additional cryptic species has been conjectured [6]. Hence, at least in part, these identifications should be revised with reference to the updated nomenclature and the use of valid DNA markers. Indeed, among the reported identifications at the species level, as little as 11 (underlined in Table 1) are reliably based on the DNA markers officially considered in *Cladosporium* taxonomy [1,2]; for the time being, this limited insight does not allow us to advance hypotheses on any definite relationships between *Cladosporium* and insect species. The prevalence of findings concerning Hemiptera and Coleoptera is evident (Figure 1), in clear connection with their higher agricultural relevance.

In addition to *C. fulvum*, which has been reclassified as *Fulvia fulva* since a long time [171], the species *C. chlorocephalum*, cited in references [18,98,172,173], has not been included since it is now synonymized with *Graphiopsis chlorocephala* following a taxonomic revision [174]. Moreover, a few cases were disregarded because the reported identification of the insect-associated fungi was clearly wrong and misleading. As an example, an isolate from the gut of the locust *Oxya chinensis* (Orthoptera, Acrididae), claimed to be *C. oxysporum* based on a 5.8S rDNA sequence [175], rather belongs to a species of *Curvularia*; this emendation unequivocally results from both a blast of the sequence in the GenBank database and a visual examination of the conidia shown in the image provided in the published paper.

With reference to the geographical distribution, there are reports from as many as 54 countries in all continents, without any implication in terms of associations possibly depending on local conditions. Especially for small-sized species, the isolations have been mainly carried out from the whole body of the insects (Figure 2); however, there are many reports considering the gut and mouth parts in relation to the possible ingestion connected to feeding habits. In this respect, isolations from frass have been considered only when this material was obtained in the laboratory without any possible interference of environmental contamination (as in references [9,21,127,136]).

The various findings of *Cladosporium* in shelters, tunnels or other structures inhabited by insects have not been exhaustively considered, since in many instances they may derive from environmental contamination rather than direct interaction with the insects. This is the case of reports of *C. cladosporioides* as the most frequent fungus found in tunnels dug by larvae of the longicorn *Rosalia alpina* (Coleoptera, Cerambycidae) in wych elm (*Ulmus glabra*) trees at a Polish conservation site [176] and in mines of lepidopteran leafminers on black locusts (*Robinia pseudacacia*) [139]. Reports on the occurrence of *Cladosporium* in food supplies of social insects (e.g., bee bread [177]) and as undesired contaminants in laboratory assays or rearing diets of insects (e.g., reference [178]) have also been disregarded.

It must be pointed out that most entries of *Cladosporium* sp. in Table 1 refer to multiple concomitant isolations or detections in the insect samples, which may imply a broad species assortment. Particularly, this connotes the many metagenomic-based studies exploring the diversity of the mycobiome of insects in specific ecological contexts. In this regard, an investigation on fungi associated with the olive fruit fly (*Bactrocera oleae*: Diptera, Tephritidae) in Calabria, Southern Italy, disclosed a striking dominance of *Cladosporium*, matching about 80% of OTU sequences. More in detail, this set was dominated by members of the *C. cladosporioides* s.c., while the *C. herbarum* s.c., *C. velox* and a couple of unidentified *Cladosporium* spp. were much less frequent [107,179]. In a similar study concerning the congeneric Queensland fruit fly (*Bactrocera tryoni*), *Cladosporium* were again among the dominant fungi in the gut mycobiome, with a higher frequency in females [108]. More recently, a systematic investigation has been carried out in the olive ecosystems of Tunisia to study the microbiome associated with this crop. However, the taxonomic remarks are reported as collective data considering the occurrence of fungi in soil and in association with insect pests, which does not allow us to infer specific associations with the latter [180]. Likewise, in an investigation concerning plant-feeding true bugs (Heteroptera: Cimicomorpha and Pentatomomorpha) carried out in China, data were summarized with reference to the family, not allowing us to infer the associations of *Cladosporium* with every single bug species [181].

Another broad-scale investigation carried out in a Canadian aspen forest showed that *Cladosporium* were the most frequent fungi associated with arthropods; again, *C. cladosporioides* was the most common species, accounting for 77% of the isolates ascribed to this genus, followed by *C. sphaerospermum* at about 20%, while *C. herbarum* and *C. orchidis* were occasional. Unfortunately, no details on the insect species representing the isolation sources were provided in this study [182].

## 3. Entomopathogenicity

Conventionally, *Cladosporium* species are not considered full-right representatives of the guild of entomopathogens, which is generally restricted to specialized fungi such as *Beauveria*, *Metarhizium* and *Lecanicillium/Akanthomyces* [183,184,185]. However, like other fungi that are widely associated with crops such as *Trichoderma* and *Talaromyces* [186,187], the evidence is increasing that *Cladosporium* may also infect insects and cause epizootics in pest populations or promote plant defense reactions.

Direct observations of the parasitic aptitude of insects are limited and essentially concern the case of *C. cladosporioides* on the sugarcane white wooly aphid (*Ceratovacuna lanigera*: Hemiptera, Aphididae); both light and electron microscopy at the host–parasite interface showed that nymphs and adults of the aphid were completely overgrown by the fungal mycelium, which penetrated and disrupted their powdery waxy coating [5].

However, circumstantial evidence of entomopathogenic aptitude in *Cladosporium* derives from several studies reporting on mortality induced by conidial suspensions administered at various concentrations and exposure times. In this regard, the available data concerning strains that proved to be effective against various targeted pests in experimental trials are summarized in Table 2.

Alternatively, the anti-insectan effect can be assessed through the addition of the fungi or their products to the laboratory diet. In this respect, when incorporated in the feed of larvae of the tobacco budworm (*Chloridea virescens*: Lepidoptera, Noctuidae), an isolate of *C. cladosporioides* was found to reduce larval and pupal weights by 56% and 7%, respectively; moreover, in preference tests, the caterpillars showed a marked tendency to avoid feed amended with the fungus [195]. Development of another noctuid moth, the tobacco cutworm (*Spodoptera litura*), was significantly prolonged when larvae were fed on a diet amended with ethyl acetate extract of *C. uredinicola* at concentrations of 1.25–2.00 μL g^−1^; moreover, at 2.00 μL g^−1^, a significantly higher number of adults emerged showing morphological deformities. At higher concentrations, significant reductions in adult emergence, longevity and reproductive potential were recorded. Finally, the toxicity of the ethyl acetate extract was further evidenced by a reduction in feed utilization by the larvae [196].

The ethyl acetate and methylene chloride extracts of a strain of *C. cladosporioides* were effective against nymphs and adults of the cotton aphid (*Aphis gossypii*: Hemiptera, Aphididae) [13,197]. Aphicidal effect was also displayed by formulations based on emulsions of culture filtrates of an endophytic strain of *C. oxysporum* endowed with proteolytic activity, which were more active than conidial suspensions against the black bean aphid (*Aphis fabae*: Hemiptera, Aphididae) [190]. In a subsequent experiment, formulations based on culture filtrates of this strain and two more endophytic isolates of *C. echinulatum* and *Cladosporium* sp. showed activity against the green peach aphid (*Myzus persicae*: Hemiptera, Aphididae), which increased at increasing concentrations. A significant reduction in the number of colonizing aphids and a relative increase in the number of winged adults were recorded. Moreover, the pretreatment of plants negatively influenced embryonic development, thus affecting fertility [198]. In the same study, consistent chitinolytic activity was determined in the culture filtrates of *Cladosporium* sp.; indeed, chitinases are considered a main factor in the bioactivity of fungal culture filtrates, as also documented for other strains of *Cladosporium* spp. [193], *C. cladosporioides* [27,48], *C. tenuissimum* and *C. xanthocromaticum* [48].

Even more, the anti-insectan effects of culture filtrates may depend on the presence of toxic compounds (Figure 3). Fungi in the genus *Cladosporium* are known as prolific producers of bioactive secondary metabolites [199], some of which have been detected as possible determinants of detrimental effects on insects. This is the case of bassianolide, a cycloligomer depsipeptide identified as a product of a strain related to the *C. cladosporioides* s.c. [200]. The alkaloid 3-(4β-hydroxy-6-pyranonyl)-5-isopropylpyrrolidin-2-one was identified in the ethyl acetate extracts of another strain of *C. cladosporioides* displaying aphicidal activity [13]. Another alkaloid, hydroxyquinoline, was identified as the potentially active product in the extracts of a strain of *C. subuliforme* [167]. The novel compound citreoviridin A was extracted from an isolate of *C. herbarum* from a marine sponge and found to inhibit the growth of larvae of the cotton leafworm (*Spodoptera littoralis*: Lepidoptera, Noctuidae) [201]. Chlorogenic acid, purified from an endophytic isolate of *C. velox*, displayed insecticidal activity by inducing significant mortality in the larvae of *S. litura* or adversely prolonging their developmental period. This phenolic compound, previously known to cause gut toxicity in lepidopterans [202], was characterized as an α-glucosidase inhibitor, performing a non-competitive type of inhibition in vitro; it also inhibited the activity of α-glycosidases in the gut of the larvae [203,204].

The importance of secondary metabolites for entomopathogenic aptitude in *Cladosporium* has been further affirmed after a study carried out on strains associated with the Chinese white wax scale (*Ericerus pela*: Hemiptera, Coccidae). This insect is known to be infected by *Cladosporium* spp. related to *C. sphaerospermum* and *C. langeronii*, which kill the scales after dramatically altering their microbiome [34]. However, the scales were later found to also harbor a non-infective *Cladosporium*. Genome sequencing showed that the non-infective strain is related to *C. cladosporioides* and has a larger genome size than a pathogenic one, which is more related to *C. sphaerospermum*. Particularly, the former has specific genes involved in nutrition pathways that are absent in the pathogen. Conversely, the latter possesses genes participating in the biosynthetic pathways of mycotoxins, such as asperfuranone, emericellamide and fumagillin. These genes were not found in the nonpathogenic strain, which, on the other hand, presented genes associated with reduced virulence [3].

### 3.1. Interactions with Biocontrol Agents

Reports on the occurrence of an association with predatory and parasitoid insects introduce the question of whether the insecticidal properties of *Cladosporium* may also affect the performances of biocontrol agents employed in crop protection. Indeed, this association can be more than merely occasional, considering that *Cladosporium* were the most abundant fungi detected in the gut of the multicolored Asian lady beetle (*Harmonia axyridis*: Coleoptera, Coccinellidae) feeding on the pea aphid (*Acyrthosiphon pisum*: Hemiptera, Aphididae) [129]. Concerning this issue, a previously mentioned strain of *Cladosporium* sp. from *H. armigera* was found not to induce significant harmful effects on a panel of beneficial predatory insects, including the red and blue beetle (*Dicranolaius bellulus*: Coleoptera, Melyridae), the transverse ladybird (*Coccinella transversalis*: Coleoptera, Coccinellidae), the green lacewing (*Mallada signatus*: Neuroptera, Chrysopidae) and the damsel bug (*Nabis kinbergii*: Hemiptera, Nabidae) [130]. Conversely, laboratory assays carried out in Egypt showed that treatment with *C. uredinicola* affected the biocontrol of the silverleaf whitefly (*Bemisia tabaci*: Hemiptera, Aleyrodidae) by the eleven-spotted ladybird (*Coccinella undecimpunctata*: Coleoptera, Coccinellidae) and the parasitoid *Eretmocerus mundus* (Hymenoptera, Aphelinidae) in various ways. In fact, all larval stages of the coccinellid were sensitive to the fungus and tended to avoid feeding on the infected whiteflies. As for the parasitoid, although mortality of the exposed individuals was low, most females avoided laying eggs on treated nymphs; nevertheless, the combined use of *C. uredinicola* and *E. mundus* was found to synergistically increase the suppression of nymphs [205].

Olfactory experiments carried out in the laboratory indicated that the parasitoid wasp *Lysiphlebus fabarum* (Hymenoptera, Braconidae) can detect cues from aphids (*A. fabae*) infected by a pathogenic strain of *Cladosporium* sp. and avoid them; hence, the employment of this strain in the field could not affect the performance of the parasitoid, implying compatibility between these, and possibly more, biological control agents of aphids [157].

### 3.2. Plant-Mediated Interactions

In addition to arising after direct contact or ingestion of conidia, the entomopathogenic effects of *Cladosporium* can also be exerted in planta, as promoted by strains able to develop endophytically. Indeed, it is known that endophytic fungi may improve plant resistance to biotic adversities through various mechanisms, including general effects on fitness and growth promotion eventually exerted in synergistic relationships with other components of the plant microbiome [206,207]. The belief is gaining ground that these valuable properties could be exploited for improving yields while reducing the input of chemicals in crop management [208,209].

Cauliflower plants artificially infected with an endophytic strain of *C. uredinicola* did not show any disease symptoms, and the vigor of endophyte-infected plants also did not differ from untreated plants. Interestingly, larvae of *S. litura* feeding on leaves from treated plants were sluggish and underwent significantly higher mortality than the control. Most of the larvae died at the time of molting to the last instar, while the survivors took a significantly longer time to pupate and further suffered significantly higher mortality at the pupal stage. In the end, fewer adults emerged from larvae on endophyte-supplemented plants; some adults exhibited morphological deformities, such as crumpled and unequal wings, and survived for a very short time. Inhibitory effects were also observed on the reproductive potential and the hatchability of eggs. The life span of females that emerged from larvae fed on plants hosting *C. uredinicola* reduced significantly, while male longevity remained unaffected [210]. All these effects were assumed to depend on physiological changes induced by the endophyte. In fact, further studies disclosed cytotoxic effects on hemocytes of *S. litura* fed on endophyte-supplemented cauliflower plants, which showed changes in shape, extensive vacuolization and necrosis. Moreover, these abnormalities increased along with the feeding duration and ultimately resulted in adverse consequences on the fitness and survival of the insect [211].

However, it is quite intuitive to consider that plant-mediated relationships should be examined case by case, as the outcome of the interaction is not necessarily unfavorable to the insects. When inoculated in perennial thistle (*Cirsium arvense*), where it is known to develop endophytically, *C. cladosporioides* increased feeding of the thistle tortoise beetle (*Cassida rubiginosa*: Coleoptera, Chrysomelidae), while it had no effect on the cabbage moth (*Mamestra brassicae*: Lepidoptera, Noctuidae). Nevertheless, dual infection with *C. cladosporioides* and *Trichoderma viride* greatly reduced beetle feeding [212]. These findings indicate that the promoting effects of *C. cladosporioides*, as well as of other endophytes, depend on both the degree of specialization of the herbivore and the species assortment in the plant microbiome, which in turn may induce chemical changes in the host. Undoubtedly, these fungi deserve higher attention in the study of insect–plant interactions, considering that their endophytic occurrence could remarkably influence insect growth and even pest population dynamics.

## 4. Other Ecological Relationships

Mycophagy somehow represents the reverse condition of entomopathogenicity, in which insects perform a suppressive role on *Cladosporium* by feeding on the mycelium. However, this relationship may still imply an ecological advantage for the fungus, deriving from the use of insects as carriers for its propagation [213]. Springtails (Collembola) are especially known for feeding on soil fungi, including *Cladosporium* [214,215], and *C. cladosporioides* has been used as feed to preserve laboratory stocks of the species *Hypogastrura tullbergi* (Poduromorpha, Hypogastruridae) and *Proisotoma minuta* (Entomobryomorpha, Isotomidae) [216]. This species was also found to support the development of the minute brown scavenger beetle (*Dienerella argus*: Coleoptera, Latridiidae) [217]; moreover, it is part of the diet of the sap beetle *Brachypeplus glaber* (Coleoptera, Nitidulidae), as demonstrated by gut content analyses and observations of adult and larval feeding [19].

Several insect pests of stored grains belonging to different orders and families have been reported to feed and even reproduce on *Cladosporium* grown in axenic cultures. More in detail, a strain of *C. cladosporioides* was found to support the development of the reticulate-winged booklouse (*Lepinotus reticulatus*: Psocoptera, Atropidae) and a series of Coleoptera, including the narrownecked grain beetle (*Anthicus floralis*: Anthicidae), the sigmoid fungus beetle (*Cryptophagus varus*: Cryptophagidae), the lesser grain borer (*Rhyzopertha dominica*: Bostrichidae), the wheat weevil (*Sitophilus granarius*: Curculionidae), the bean weevil (*Acanthoscelides obtectus*: Bruchidae), the larger black flour beetle (*Cynaeus angustus*: Tenebrionidae), the square-nosed fungus beetle (*Lathridius minutus*) and *Microgramma arga* (Lathrididae) [218]. One of them, *C. angustus*, was even attracted by fresh-milled corn flour amended with *Cladosporium* [219]. Moreover, the ability to develop in vitro on pure cultures of an unidentified *Cladosporium* strain was reported for the foreign grain beetle (*Ahasversus advena*: Coleoptera, Sylvanidae), which females also demonstrated to prefer *Cladosporium* in oviposition tests [220].

Some contrast is evident in reports concerning the interaction of *C. cladosporioides* with *R. dominica*. In fact, after the above supporting effect [218], induction of mortality has recently been documented in this borer insect [56]. This contradiction could be easily explained considering that different strains of the same species may have different biological properties, particularly with reference to secondary metabolite production. Moreover, after the recent adjustments in *Cladosporium* taxonomy, it is quite possible that, actually, these isolates belong to different species, which further underlines the importance of a correct identification. Rather than being a mere hypothesis, this inference was verified in the case of the *Cladosporium* associates of the fruit fly *Drosophila suzukii* (Diptera, Drosophilidae) in raspberries; in fact, it transpired that isolates preliminarily identified as *C. cladosporioides* belonged to at least two more species (*C. anthropophilum* and *C. pseudocladosporioides*) after a more accurate taxonomic identification based on biomolecular markers [9].

The ambrosia beetles (Coleoptera, Curculionidae, Scolytinae) are known to spread fungi that develop in the galleries they dig in the host trees [221]. Although experimental evidence leans for selected fungi, such as species of *Fusarium*, *Geosmithia*, *Penicillium* and *Raffaelea*, to be more systematically involved in these mutualistic relationships [36,52,222,223], *Cladosporium* is often isolated both from the insects and their galleries [52,94,133,134]. Particularly, isolation and identification based on biomolecular methods demonstrated the association of *C. perangustum* with *Xylosandrus compactus* after female beetles were found carrying the fungus on their body [77]; previously, *Cladosporium* had been regarded as the principal food source of this species [224].

Likewise, the coffee berry borer (*Hypothenemus hampei*: Coleoptera, Curculionidae) was found to be associated with *C. oxysporum* and an unidentified *Cladosporium* sp. [76]. Another unidentified *Cladosporium* sp. was found to systematically occur in leaf rolls inhabited by larvae of the weevil *Euops lespedezae* (Coleoptera, Attelabidae) on the leafy bush clover (*Lespedeza cyrtobotrya*); however, this fungus was not found in mycangia of the weevil females, indicating that it is not a vertically transmitted symbiont unlike other leaf roll-associated fungi [225]. *Cladosporium* spp. were also dominant in leaf rolls of another Attelabidae, *Heterapoderopsis bicallosicollis*, on the Chinese tallow tree (*Triadica sebifera*); this is considered not to be a specific association, as the authors hypothesize that these and other cellulolytic fungi, providing nutritional support to the larvae, colonize the leaf rolls once they fall to the soil [226].

In addition to the above findings, not surprisingly, our fungi have been reported to colonize galls formed on plants by insects of various taxonomic assortment. This is the case of aphids of the genus *Pemphigus* (Hemiptera, Pemphigidae), commonly associated with poplars (*Populus* spp.), in which galls *C. cladosporioides* and other unidentified *Cladosporium* spp. can be found among other fungi [227]. Moreover, *C. sphaerospermum* was found to frequently occur in galls of the eastern spruce adelgid (*Adelges abietis*: Hemiptera, Adelgidae), even if no circumstantial relationship could be documented [228]. Also very common is the occurrence of *Cladosporium* in galls formed on a multitude of plants by Asphondylia spp. and other midges in the Cecidomyidae (Diptera) [103,104,105,132,229]. Again, this association does not seem to entail any functional symbiotic relationships, considering that on Greek savory (Micromeria graeca), these fungi were also found in ungalled flower buds. Moreover, the isolates did not belong to a single species, which could have implied an ecological specialization; rather, they were representative of a wide taxonomic assortment, including two novel species [16].

Connected with the feeding aptitude is the isolation of *Cladosporium* spp. from the gut of larvae of aquatic shredders of the genera *Phylloicus* (Trichoptera, Calamoceratidae) and *Stenochironomus* (Diptera, Chironomidae); these strains displayed cellulolytic and xylanolytic properties in laboratory assays, supporting the hypothesis that they might improve the digestibility of leaves by the insects [63,73,150]. Likewise, *Cladosporium* spp. were reported as prevalent in the gut of males of the Asian tiger mosquito (*Aedes albopictus*: Diptera, Culicidae) and are considered to play a role in the assimilation of fructose, which is a relevant component in their nectarivorous diet [86]. As part of an extensive array of microorganisms that play a crucial role in the digestion of feed, in the absorption of nutrients and in the protection against pathogens, the occurrence of *Cladosporium* in the digestive tract has been documented in many unrelated insect species, both after direct isolation [3,15,19,24,26,47,50,51,58,59,78,79,100,101,118,125,127] and as a result of studies based on biochemical (e.g., denaturing gradient gel electrophoresis) and metagenomic analyses [3,40,47,88,90,101,107,108,109,117,118,125,133,138,141,144,146,151,154,161,230]. In addition to descriptive aims, these studies have addressed various aspects that more or less influence the gut microbiome species assortment, such as age, instar, gender, caste, diet, pesticides, antibiotics and various environmental factors. Increasing evidence indicates that disturbances in the gut microbiota can compromise the host’s health and that its diversity has a far-reaching impact on the insect’s fitness. This is also closely related to the issue of pesticide resistance; in fact, the capacity to degrade chlorpyrifos and endosulfan has been documented by strains of *C. cladosporioides* [231,232,233]. Indeed, a better understanding of the role played by these fungi and the interacting microbial communities in the insects’ gut is essential in view of increasing opportunities for their possible exploitation [230,234,235].

The helper role of *Cladosporium* towards pests can be reciprocated in the cases where insects act as vectors of plant pathogenic strains. In this respect, *D. suzukii* was found to act as a carrier of *Cladosporium* spp., enabling these fungi to proliferate in raspberries [9,127]. The dark-winged fungus gnat *Bradysia impatiens* (=*B. agrestis*: Diptera, Sciaridae) has also been reported to spread *Cladosporium* conidia [78]. Clearly, the possible epidemiological role of some insects as occasional vectors of plant pathogenic species must be taken into careful consideration, even if the impact of *Cladosporium* spp. as plant pathogens is generally lower than other insect-associated fungi such as *Fusarium* [236].

A particular kind of interaction with *Cladosporium* has been documented for two wasps (*Dolichovespula sylvestris* and *Paravespula vulgaris*: Hymenoptera, Vespidae) that act as pollinators for the orchid *Epipactis helleborine*. In fact, the wasps were found to spread conidia of *Cladosporium* while visiting the flowers; these fungi are presumed to produce ethanol contained in the nectar, which in turn is thought to attract the wasps [126]. This finding could be indicative of a more widespread and relevant role of *Cladosporium* in the interaction with pollen vectors for angiosperms.

*Cypripedium fargesii* is a nectarless endangered orchid that requires cross-pollination to produce the maximum number of viable embryos. The flat-footed fly (*Agathomyia* sp.: Diptera, Platypezidae) is used to visit flowers of this orchid, and individuals examined after entering or leaving the labellum sac were found to carry *Cladosporium* conidia on their legs and mouthparts, suggesting mycophagy. Quite intriguingly, the upper surface of foliage presents blackish hairy spots that mimic the mold spots of *Cladosporium*, thereby serving as visual lures. Moreover, the floral scent composition includes some volatile compounds, such as 3-methyl-1-butanol, 2-ethyl-1-hexanol and 1-hexanol, which have also been detected in *Cladosporium* cultures [89]. Based on these remarks, it can be inferred that the endophytic adaptation of *Cladosporium* has led this fungus to play a fundamental intermediary role in the reproduction of these plants, which deserves to be further investigated.

## 5. Conclusions

Within the diverse ecological relationships connecting insects and fungi [237], evidence from the available experimental data is mostly indicative of an antagonistic role of *Cladosporium* against insects deserving higher attention for practical exploitation in pest control. In this respect, the reported infectious aptitude of *C. cladosporioides* against spider mites [238,239] and the known mycoparasitic ability against several plant pathogens, such as rusts [240], candidate these fungi as multipurpose biocontrol agents. Clearly, this opportunity requires an assessment of whether these useful roles can be simultaneously performed by single selected strains. The recent observation that phylogenetically diverse species of *Cladosporium* display concurrent effects supports the conjecture that these functional traits are rooted in ancestry in this genus, providing a favorable indication in this respect [241].

A final aspect connected with the *Cladosporium*–insect association to be taken into careful consideration refers to the recent trend of introducing insects into the human diet [242,243]. Indeed, the high frequency with which these fungi can be found infecting or coating insects should be particularly monitored in the rearing conditions, which can be conducive to their spread and the ensuing possible mycotoxin contamination [199,244,245,246].

## Figures and Tables

**Figure 1 jof-10-00078-f001:**
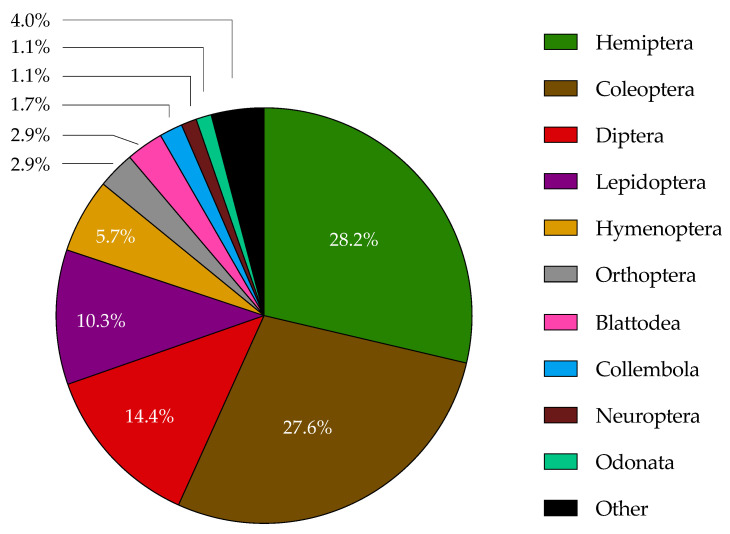
Findings of insect-associated *Cladosporium* spp. grouped by insect orders.

**Figure 2 jof-10-00078-f002:**
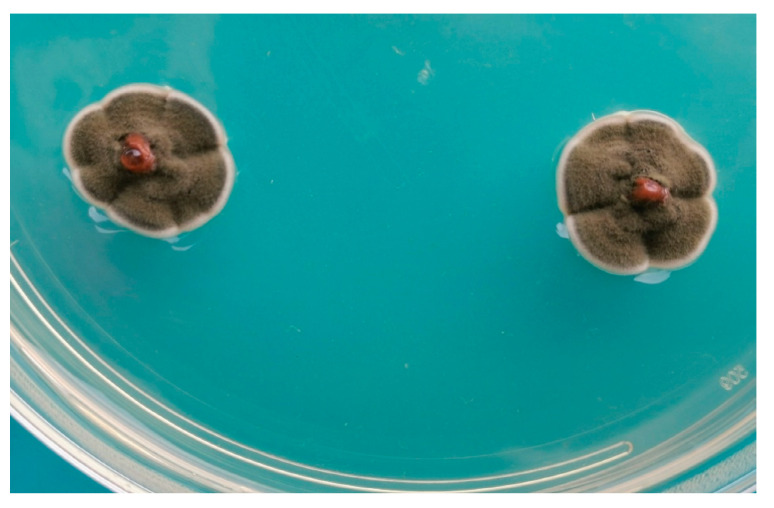
Isolation of *Cladosporium* from the pine tortoise scale (*Toumeyella parvicornis*).

**Figure 3 jof-10-00078-f003:**
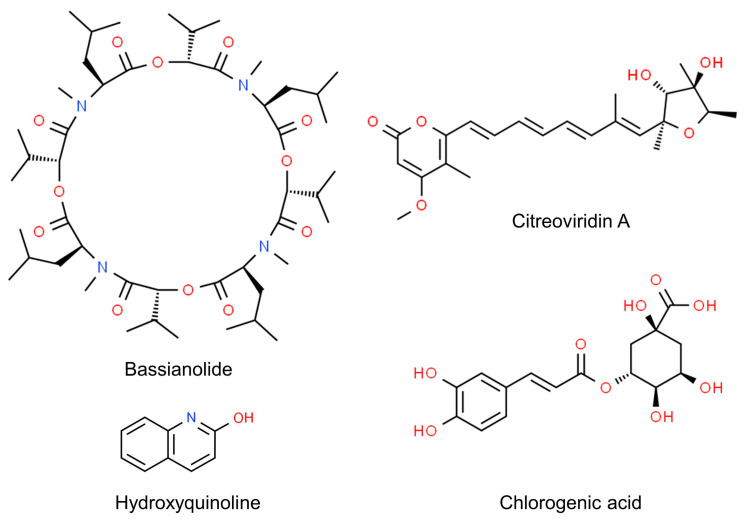
Chemical structure of *Cladosporium* secondary metabolites displaying anti-insectan effects.

**Table 1 jof-10-00078-t001:** Occurrence of *Cladosporium* in association with insect species.

*Cladosporium* Species	Insect Species *	Location	Reference
*C. aggregatocicatricatum*	* Dryocosmus kuriphilus *	Monti Cimini (Italy)	[7]
	* Xylosandrus compactus *	Circeo Promontory (Italy)	[8]
*C. anthropophilum*	* Drosophila suzukii *	Maryland (USA)	[9]
*C. aphidis*	unidentified aphid	Piracicaba (São Paulo, Brazil) Berlin; Brandenburg (Germany) Parma (Italy) Wilmington (Delaware, USA)	[1]
	* Aphis gossypii *	Puerto Rico	
	* Aphis * sp.	Berlin (Germany)	
	* Aphis symphyti *	Klosterneuburg (Austria)	
	* Brevicoryne brassicae *	Assiut (Egypt)	[10]
	* Rhopalosiphum maidis *	Honolulu (Hawaii, USA)	[1]
	unidentified scale	Essen (Germany)	
*C. austrohemisphaericum*	* X. compactus *	Circeo Promontory (Italy)	[8]
*C. chasmanthicola*	* Spodoptera frugiperda *	Kansas (USA)	[11]
*C. cladosporioides*	* Aleurothrixus aepim *	Brazil	[12]
	* Aphis craccivora *	Egypt	[13]
	* Apion ulicis *	Christchurch area (New Zealand)	[14]
	* Apis mellifera *	Assiut Governorate (Egypt)	[15]
	* Asphondylia micromeriae * * Asphondylia nepetae *	Isle of Vivara (Italy) Averno; Matera (Italy)	[16]
	* Atta capiguara * * Atta laevigata *	Botucatu (São Paulo, Brazil)	[17]
	* Bemisia * spp.	Dakahlia Governorate (Egypt)	[18]
	* Brachypeplus glaber *	Florida (USA)	[19]
	* B. brassicae *	Dakahlia Governorate (Egypt) Assiut (Egypt)	[20] [10]
	* Ceratovacuna lanigera *	Anakapalle (Andhra Pradesh, India)	[5]
	* Chitaura brachyptera *	Dumoga-Bone (Sulawesi, Indonesia)	[21]
	* Chrysomphalus aonidum *	Qualubia (Egypt)	[22]
	* Chrysomya megacephala *	Ludhiana (Punjab, India)	[23]
	* Chrysoperla rufilabris *	Monroe; Oktibbeha (Mississippi, USA)	[24]
	unidentified lignicolous Coleoptera	Ås (Norway)	[25]
	* Coptotermes formosanus *	New Orleans (Louisiana, USA)	[26]
	* Culex pipiens *	Qena Governorate (Egypt)	[27]
	* Culex quinquefasciatus *	Basrah (Iraq)	[28]
	* Cydia ulicetana *	Christchurch area (New Zealand)	[14]
	* Desoria albella *	Warren Woods (Michigan, USA)	[29]
	* Diaphorina citri *	Florida (USA)	[30]
	unidentified Diptera	Coimbra (Portugal)	[31]
	* D. suzukii *	Maryland (USA) Geneva (New York, USA)	[9] [32]
	* Dytiscus marginalis *	East Anatolia (Turkey)	[33]
	* Epiphyas postvittana *	Christchurch area (New Zealand)	[14]
	* Ericerus pela *	Kunming (China)	[34]
	* Euphalerus clitoriae *	Pernambuco (Brazil)	[35]
	* Euwallacea interjectus *	Hiroshima Prefecture (Japan)	[36]
	* Galleria mellonella *	Northern China Beijing (China)	[37] [38]
	* Gromphadorhina portentosa *	Columbus (Ohio, USA)	[39]
	unspecified grasshoppers	Ulu-Endau (Malaysia)	[21]
	* Hermetia illucens *	Giessen (Germany)	[40]
	* Heteraphorura subtenuis *	Alberta (Canada)	[41]
	* Homalodisca vitripennis *	Southern California (USA)	[42]
	* Hydrophilus piceus *	East Anatolia (Turkey)	[33]
	unidentified dead insect	Thailand	[43]
	* Ips sexdentatus *	Northeastern Ukraine	[44]
	* Kermes * sp.	Guangdong (China)	[45]
	* Lycorma delicatula *	Berks County (Pennsylvania, USA)	[46]
	* Megaplatypus mutatus *	Bragado (Argentina)	[47]
	* Mesambria maculipes *	Dumoga-Bone (Sulawesi, Indonesia)	[21]
	* Myzus persicae *	Salah El-Din Governorate (Iraq)	[48]
	* Nilaparvata lugens *	Gazipur (Bangladesh)	[49]
	* Odontotermes formosanus *	Jinhua (China)	[50]
	* Pantala flavescens *	Jinhua (China)	[51]
	* Pityogenes bidentatus *	Babimost; Mielec; Opole (Poland)	[52]
	* Prays oleae *	Mirandela-Bragança region (Portugal)	[53]
	* Pterostichus melanarius *	Assiut Governorate (Egypt)	[15]
	unidentified Pyrrhocoridae	Lebanon	[54]
	* Rhynchophorus ferrugineus *	Assiut Governorate (Egypt)	[15]
	* Scolytogenes birosimensis *	Central and Western Japan	[55]
	* Sericothrips staphylinus *	Christchurch area (New Zealand)	[14]
	* Sitophilus oryzae *	Multan (Pakistan)	[56]
	unidentified Thysanura	Coimbra (Portugal)	[31]
	* Tomicus piniperda *	Mościska Forest (Poland)	[57]
	* Triatoma brasiliensis * * Triatoma pseudomaculata *	Rio de Janeiro (Brazil)	[58]
	* Triatoma infestans *	Mendoza; Salta; Santa Fe (Argentina)	[59]
	* Troglophilus neglectus *	Tolmin (Slovenia)	[60]
*C. cucumerinum*	unidentified Diptera unidentified Thysanura	Coimbra (Portugal)	[31]
*C. cycadicola*	* Tribolium castaneum *	Iksan (South Korea)	[61]
*C. delicatulum*	* Thrips * sp.	Hamedan (Iran)	[62]
*C. dominicanum*	* X. compactus *	Circeo Promontory (Italy)	[8]
*C. endophyticum*	* Stenochironomus * sp.	Adolpho Ducke Reserve (Amazonas); Lajeado State Park (Tocantins, Brazil)	[63]
*C. exasperatum*	* Nasutitermes octopilis *	Nouragues Nature Reserve (French Guiana)	[64]
*C. exile*	* Aphis * sp.	Somesara (Iran)	[62]
*C. halotolerans*	* G. mellonella *	Napoli area (Italy)	[65]
	* Stenochironomus * sp.	Lajeado State Park (Tocantins, Brazil)	[63]
	* Thitarodes xiaojinensis *	Xiaojin (China)	[66]
	* T. castaneum *	Sangju (South Korea)	[61]
*C. herbarum*	* Aleurodicus cocois *	Recife (Brazil)	[67]
	* A. gossypii *	Latvia	[68]
	* A. ulicis *	Christchurch area (New Zealand)	[14]
	* A. mellifera *	Assiut Governorate (Egypt)	[15]
	* B. brassicae *	Assiut (Egypt)	[10]
	* C. ulicetana *	Christchurch area (New Zealand)	[14]
	* D. marginalis *	East Anatolia (Turkey)	[33]
	* E. postvittana *	Christchurch area (New Zealand)	[14]
	* H. subtenuis *	Alberta (Canada)	[41]
	* H. piceus *	East Anatolia (Turkey)	[33]
	* Musca domestica *	Seropedica (Rio de Janeiro, Brazil)	[69]
		Gauteng Province (South Africa)	[70]
	* P. bidentatus *	Babimost (Poland)	[52]
	* Pseudopsis subulata *	Montmorency Forest (Canada)	[71]
	* S. staphylinus *	Christchurch area (New Zealand)	[14]
	* Tenebrio molitor *	Aarhus (Denmark)	[72]
	* T. piniperda *	Mościska Forest (Poland)	[57]
	* T. brasiliensis * * T. pseudomaculata *	Rio de Janeiro (Brazil)	[58]
*C. iranicum*	unidentified scale	Iran	[1]
*C. kenpeggii*	* Stenochironomus * sp.	Adolpho Ducke Reserve (Amazonas); Lajeado State Park (Tocantins, Brazil)	[63]
	* Triplectides * sp.	Lajeado State Park (Tocantins, Brazil)	[73]
*C. langeronii*	* D. kuriphilus *	Monti Cimini (Italy)	[7]
	* E. pela *	Kunming (China)	[34]
*C. macrocarpum*	* H. subtenuis *	Alberta (Canada)	[41]
*C. oxysporum*	* Anoplolepis custodiens * * Aonidiella aurantii *	Eastern Transvaal (South Africa)	[74]
	* A. gossypii *	Mataffin (South Africa)	[75]
	* A. mellifera *	Assiut Governorate (Egypt)	[15]
	* C. lanigera *	Anakapalle (Andhra Pradesh, India)	[5]
	* C. aonidum *	Eastern Transvaal (South Africa)	[74]
	* Hypothenemus hampei *	El Tizal (Mexico)	[76]
	unidentified Muscidae	Eastern Transvaal (South Africa)	[74]
	* Planococcus citri *	Mataffin (South Africa)	[75]
	* P. oleae *	Mirandela–Bragança region (Portugal)	[53]
	* Pseudococcus longispinus *	Eastern Transvaal (South Africa)	[74]
	* P. melanarius *	Assiut Governorate (Egypt)	[15]
	* Pulvinaria aethiopica * * Toxoptera citricidus * * Trioza erytreae *	Eastern Transvaal (South Africa)	[74]
*C. perangustum*	* X. compactus *	Grottammare (Italy)	[77]
*C. pini-ponderosae*	unidentified Diptera	Coimbra (Portugal)	[31]
*C. pseudocladosporioides*	* Eulophid parasitoid *	Averno; Napoli (Italy)	[16]
	* D. suzukii *	Maryland (USA)	[9]
*C. ramotenellum*	* A. nepetae *	Napoli (Italy)	[16]
*C. sphaerospermum*	* Bradysia impatiens *	South Korea	[78]
	* B. brassicae *	Assiut (Egypt)	[10]
	* D. kuriphilus *	Monti Cimini (Italy)	[7]
	* E. pela *	Kunming (China)	[34]
	unidentified grasshoppers	Ulu-Endau (Malaysia)	[21]
	* H. vitripennis *	Southern California (USA)	[42]
	* H. subtenuis *	Alberta (Canada)	[41]
	* Imbrasia belina *	Francistown (Botswana)	[79]
	* M. maculipes *	Dumoga-Bone (Sulawesi, Indonesia)	[21]
	* Macrotermes barneyi *	Guangdong (China)	[80]
	* Onychiurus pseudofimetarius *	Auckland (New Zealand)	[1]
	* P. melanarius * * R. ferrugineus *	Assiut Governorate (Egypt)	[15]
	* S. birosimensis *	Central and Western Japan	[55]
	* T. piniperda *	Mościska Forest (Poland)	[57]
	* T. infestans *	Mendoza; Santa Fe (Argentina)	[59]
	* T. castaneum *	Goseong; Sangju (South Korea)	[61]
*Cladosporium* sp.	* Acmaedora flavolineata *	Turkey	[81]
	* Acronicta major *	Hangzhou (China)	[82]
	* Adelges piceae *	Gaspé Peninsula (Canada)	[83]
	* Adelges tsugae *	Northeastern USA	[84]
	* Aedes albopictus *	Nice; Portes-lès-Valence; Saint Priest (France) Mananjary; Toamasina; Tsimbazaza (Madagascar) Bình Dương; Hồ Chí Minh; Vũng Tàu (Vietnam) Villeurbanne (France) Lubbock (Texas, USA)	[85] [86] [87]
	* Aedes aegypti *	Lubbock (Texas, USA)	[87]
	* Aedes japonicus *	Urbana (Illinois, USA)	[88]
	* Aedes triseriatus *		
	* Agathomyia * sp.	Yaoshan Mountain (China)	[89]
	* Agrilus mali *	Yining (China)	[90]
	Aleocharinae spp.	Denmark	[91]
	* Aleurocanthus spiniferus *	Southern Anhui (China)	[92]
	* A. aepim *	Bahia (Brazil)	[93]
	* Alniphagus aspericollis *	Greater Vancouver region (Canada)	[94]
	* Anastrepha fraterculus *	Caxias do Sul (Rio Grande do Sul, Brazil)	[95]
	* Anopheles coluzzii *	Koulikoro (Mali)	[96]
	* Apertochrysa formosanus *	Sugadaira-Kogen (Japan)	[97]
	* A. craccivora * * Aphis durantae * * A. gossypii *	Dakahlia Governorate (Egypt)	[98]
	* Apis cerana *	Chiang Mai (Thailand)	[99]
	* A. mellifera *	Tucson (Arizona, USA) Assiut Governorate (Egypt) Grugliasco (Italy)	[100] [15] [101]
		Halle (Germany) Athens (Greece) Lublin (Poland) Chiang Mai (Thailand) London (United Kingdom)	[102] [99]
	* Asphondylia glabrigerminis *	Mittagong; Melbourne area (Australia)	[103]
	* A. micromeriae *	Astroni Nature Reserve (Italy)	[104]
	* Asphondylia serpylli *	Lublin area (Poland)	[105]
	* A. capiguara * * A. laevigata *	Botucatu (São Paulo, Brazil)	[17,106]
	* Atomaria * spp.	Denmark	[91]
	* Bactrocera oleae *	Calabria (Italy)	[107]
	* Bactrocera tryoni *	New South Wales; Victoria (Australia)	[108]
	* Bemisia tabaci *	Dakahlia Governorate (Egypt)	[98]
	* Bombus terrestris *	Beijing (China)	[109]
	* Bombyx mori *	Hangzhou (China)	[82]
	* B. impatiens *	South Korea	[110]
	* Calliopum aeneum *	Denmark	[91]
	* Callosobruchus maculatus *	Columbus; Miami County (Ohio); Erie County (Pennsylvania, USA)	[111]
	Carabidae spp.	Uckermark (Germany)	[112]
	* Carpophilus * sp.	Gualmatán (Colombia)	[113]
	* Ceroplastes floridensis *	Mansoura (Egypt) Kiryat Tivon (Israel)	[22] [114]
	* Ceroplastes rusci *	Mansoura (Egypt) Larnaca (Cyprus)	[115] [114]
	* Ceroplastes * sp.	Malaga (Spain)	[114]
	* Chalcophora detrita *	Turkey	[116]
	* Coccus hesperidum *	Ramat HaShofet (Israel)	[114]
	unspecified Coleoptera	Ås (Norway)	[25]
	unidentified Collembola	Warren Woods (Michigan, USA)	[29]
	* Colletes cunicularius *	Ave-et-Auffe (Belgium)	[117]
	* Cortinicara gibbosa *	Denmark	[91]
	* Crocothemis servilia *	Hefei (China)	[118]
	* Ctenolepisma longicaudatum *	Wien (Austria)	[119]
	* C. pipiens *	San Francisco; San Rafael (California, USA)	[120]
	* C. quinquefasciatus *	Nakhon Nayok (Thailand) Mar del Plata (Argentina)	[121] [122]
	* Cydia pomonella *	Austria	[123]
	* Cytilus sericeus *	Ostrava (Czechia)	[124]
	* Diabrotica * sp.	Gualmatán (Colombia)	[113]
	* Diabrotica virgifera *	Deutsch-Jahrndorf (Germany)	[125]
	* Diaphania pyloalis *	Hangzhou (China)	[82]
	unidentified Diptera	Warren Woods (Michigan, USA)	[29]
	* Dolichovespula sylvestris *	Havreballe Forest (Denmark)	[126]
	* D. suzukii *	Maryland (USA)	[127]
	Drosophilidae spp. * Enicmus transversus *	Denmark	[91]
	* E. pela *	Kunming (China)	[3]
	* Ferrisia virgata *	Madurai (India)	[128]
	* H. vitripennis *	Southern California (USA)	[42]
	* Harmonia axyridis *	Hubei (China)	[129]
	* Helicoverpa armigera *	Narrabri; New South Wales (Australia)	[130]
	* H. hampei *	Chiapas (Mexico) El Tizal (Mexico)	[131] [76]
	* Illiciomyia yukawai *	Mie Prefecture (Japan)	[132]
	* Ips acuminatus *	Libavá (Czechia)	[133]
	* Ips cembrae * * Ips duplicatus *	Rouchovany (Czechia)	
	* I. sexdentatus *	Rouchovany (Czechia) Northeastern Ukraine	[133] [44]
	* Ips typographus *	Rouchovany (Czechia)	[133]
	unidentified Lepidoptera	Coimbra (Portugal)	[31]
	* Liparthrum colchicum *	Migliarino Natural Park (Italy)	[134]
	* Loberus impressus *	Iberia Parish (Louisiana, USA)	[135]
	* Lonchodes brevipes *	Singapore	[136]
	* Lonchoptera * spp.	Denmark	[91]
	* Lutzomyia * sp.	Antioquia Department (Colombia)	[137]
	* Lymantria dispar asiatica *	Harbin (China)	[138]
	* Macrosaccus robiniella *	Transylvania (Romania)	[139]
	* Marchalina hellenica *	Ischia (Italy)	†
	* Matsumurasca onukii *	various regions in China	[140]
	* Meiltaea cinxia *	Eckerö; Sund (Finland)	[141]
	* Milviscutulus mangiferae *	Upper Galilee (Israel)	[114]
	*Minettia fasciata* *Minettia longipennis* *Minettia plumicornis*	Denmark	[91]
	* M. domestica *	Ahwaz (Iran)	[142]
		Gauteng Province (South Africa)	[70]
		Bruxelles (Belgium) Butare (Rwanda)	[143]
	* Neobathyscia mancinii * * Neobathyscia pasai *	Damati Cave (Italy) Tana delle Sponde Cave (Italy)	[144]
	* N. lugens *	Hangzhou (China)	[145]
	* Orchelimum vulgare *	Houston (Texas, USA)	[146]
	* Orthoperus brunnipes *	Denmark	[91]
	* Orthotomicus erosus *	Italian harbors, on imported wood	[147]
	* Ostrinia nubilalis *	Andau (Germany)	[125]
	* Paracoccus marginatus *	Madurai (India)	[128]
	* Parasaissetia nigra *	Kiryat Tivon (Israel)	[114]
	* Paravespula vulgaris *	Havreballe Forest (Denmark)	[126]
	* Parectopa robiniella *	Transylvania (Romania)	[139]
	* Phlebotomus papatasi *	Tehran (Iran)	[148]
	* Phlebotomus * spp.	Tunisia	[149]
	* Phylloicus * sp.	Adolpho Ducke Reserve (Amazonas, Brazil)	[150]
	* Pieris brassicae *	Bornem (Belgium) Randwijk; Wageningen (Netherlands)	[151]
	* Planococcus ficus *	Gholan Heights (Israel)	[152]
	* Poecilocerus pictus *	Chennai (India)	[153]
	* Polygraphus polygraphus *	Rouchovany (Czechia)	[133]
	* Probergrothius angolensis *	Namibia	[154]
	* Pseudatomoscelis seriatus *	College Station (Texas, USA)	[155]
	unidentified Psocoptera	Warren Woods (Michigan, USA)	[29]
	* Psylliodes attenuata *	Daqing; Harbin; Changchun; Qujing (China)	[156]
	* P. oleae *	Mirandela–Bragança region (Portugal)	[53]
	* P. melanarius *	Assiut Governorate (Egypt)	[15]
	* Pulvinaria aurantii *	Northern Iran	[157]
	* Pulvinaria psidii *	Mansoura (Egypt)	[22]
	* Pulvinaria tenuivalvata *	Dakahlia Governorate (Egypt)	[158]
	* R. ferrugineus *	Assiut Governorate (Egypt)	[15]
	* Saissetia * sp.	Larnaca (Cyprus)	[114]
	* Sapromyza quadripunctata *	Denmark	[91]
	* Scolypopa australis *	Nelson (New Zealand)	[159]
	* S. birosimensis *	Central and Western Japan	[55]
	* S. frugiperda *	Guangzhou (China) Kansas (USA)	[160] [12]
	* Stenochironomus * sp.	Adolpho Ducke Reserve (Amazonas); Lajeado State Park (Tocantins, Brazil)	[63,150]
	* Stephostethus lardarius * * Stilbus testaceus *	Denmark	[91]
	* Taeniothrips inconsequens *	Northeastern USA	[84]
	* T. molitor *	Aarhus (Denmark) Shenyang (China)	[72] [161]
	* Thaumastocoris peregrinus *	Southeast Uruguay	[162]
	unidentified Thysanura	Coimbra (Portugal)	[31]
	* Toumeyella parvicornis *	Campania (Italy)	[163]
	* Trialeurodes ricini *	Dakahlia Governorate (Egypt)	[98]
	* T. brasiliensis * * T. pseudomaculata *	Rio de Janeiro (Brazil)	[58]
	* T. castaneum *	South Korea	[61]
	* Triplectides * sp.	Lajeado State Park (Tocantins, Brazil)	[73]
	* Xyleborinus saxesenii *	Italian harbors, on imported wood Northeastern Italy South Florida (USA)	[147] [164] [165]
	* Xyleborus bispinatus * * Xyleborus volvulus *	South Florida (USA)	[165]
	* X. compactus *	Central Florida (USA) Migliarino Natural Park (Italy)	[166] [134]
	* Xylosandrus crassiusculus *	South Florida (USA)	[165]
	* Xylosandrus germanus *	Northeastern Italy	[164]
*C. subtilissimum*	* A. capiguara *	Botucatu (São Paulo, Brazil)	[17]
*C. subuliforme*	* D. citri *	Guilin (China)	[167]
	unidentified Streblidae	Furna do Morcego (Pernambuco, Brazil)	[168]
	* Triplectides * sp.	Lajeado State Park (Tocantins, Brazil)	[73]
*C. tenuissimum*	* M. mutatus *	Bragado (Argentina)	[47]
	* M. persicae *	Salah El-Din Governorate (Iraq)	[48]
	unidentified Thysanura	Coimbra (Portugal)	[31]
	* Trachymela sloanei *	Guangdong (China)	[169]
*C. uredinicola*	* A. gossypii *	Dakahlia Governorate (Egypt)	[170]
	* Bemisia * spp.	Dakahlia Governorate (Egypt)	[18]
	* B. tabaci *	Dakahlia Governorate (Egypt)	[170]
*C. velox*	* B. oleae *	Calabria (Italy)	[107]
	* T. castaneum *	Mungyeong; Sangju (South Korea)	[61]
*C. verrucocladosporioides*	* Triplectides * sp.	Lajeado State Park (Tocantins, Brazil)	[73]
*C. xanthocromaticum*	* M. persicae *	Salah El-Din Governorate (Iraq)	[48]

* Colors are indicative of the order to which the species belong, as follows: Blattodea, Coleoptera, Collembola (Entomobryomorpha, Poduromorpha), Diptera, Hemiptera, Hymenoptera, Lepidoptera, Neuroptera, Odonata, Orthoptera, Phasmatodea, Psocoptera, Thysanoptera, Thysanura, Trichoptera. Identification of the strains isolated from the underlined sources is based on complete set of DNA markers. † Original isolation recently obtained by E.R.

**Table 2 jof-10-00078-t002:** Reported effectiveness of conidial suspensions of *Cladosporium* in inducing mortality on insect pests.

*Cladosporium* Species	Source	Insect Targets	Country	Reference
*C. cladosporioides*	*Bemisia* sp.	*Bemisia* sp.	Egypt	[18]
	*Brevicoryne brassicae*	*B. brassicae*	Egypt	[20]
	*Culex quinquefasciatus*	*C. quinquefasciatus*	Iraq	[28]
	endophytic	*Duponchelia fovealis*	Brazil	[188]
	* Kermes * sp.	* Hemiberlesia pitysophila *	China	[45]
	*Lycorma delicatula*	*Tenebrio molitor*	USA	[46]
	*Myzus persicae*	*M. persicae*	Iraq	[48]
	*Nilaparvata lugens*	*Bemisia tabaci*	Bangladesh	[49]
	*Pulvinaria aurantii*	*Aphis fabae*	Iran	[157]
	*Sitophilus oryzae*	*Rhyzopertha dominica* *Sitophilus zeamais* *Trogoderma granarium*	Pakistan	[56]
	soil	*Metopolophium dirhodum*	Egypt	[189]
*C. oxysporum*	endophytic	*A. fabae*	Algeria	[190]
	endophytic	*Chilo partellus*	India	[191]
	*Planococcus citri*	*Pseudococcus longispinus* *Pulvinaria aethiopica* *Toxoptera citricida* *Trioza erytreae*	South Africa	[75]
	unknown	*Aphis craccivora*	India	[192]
*C. sphaerospermum*	endophytic	*D. fovealis*	Brazil	[188]
*Cladosporium* sp.	*Helicoverpa armigera*	*Aphis gossypii* *B. tabaci* *H. armigera*	Australia	[130]
	*Spodoptera frugiperda*	*S. frugiperda*	China	[169]
*Cladosporium* spp.	several species of sap-sucking Hemiptera	*A. craccivora* *A. gossypii* *B. tabaci*	Egypt	[98,193]
*C. subuliforme*	*Diaphorina citri*	*D. citri*	China	[167]
*C. tenuissimum*	*M. persicae*	*M. persicae*	Iraq	[48]
	*Trachymela sloanei*	*S. frugiperda*	China	[194]
*C. uredinicola*	*A. gossypii* *B. tabaci*	*A. gossypii* *B. tabaci*	Egypt	[170]
	*Bemisia* sp.	*Bemisia* sp.	Egypt	[18]
*C. xanthocromaticum*	*M. persicae*	*M. persicae*	Iraq	[48]

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
