# Peer review of "Cladosporium—Insect Relationships"

_jof, 2024, doi:10.3390/jof10010078_

Round 1
Reviewer 1 Report
Comments and Suggestions for Authors
The authors reviewed Cladosporium and insect relationships in this paper. It merits publication. Some issues I list below the authors may consider to be addressed.
1. Cladosporium distributed in all kinds of environments and were found almost in all kinds of living organisms. They were also found in the guts of insects. The authors reviewed the relationships between Cladosporium and insects mainly based on entomopathogenicity, and showed the antagonistic role of Cladosporium against hosts. However, there is little discussion (or a separate chapter) on the mutualism with insect hosts. Why insects harbor Cladosporium in the guts? It will prove more idea for related research if the authors add some information or viewpoints on the adaptation of Cladosporium to hosts and evolution of mutualism relationship with insects.
2. Table 1
The authors indicated different Orders with different colors in the table. However, it was inconvenience to find the Order that the insects belong. Maybe it will be better to add a column with Orders in the table and reorder the insect species belong the same Order in each Cladosporium species line.
3. Figure 2.
It’s puzzling why the authors put this figure. There is no culture process and identification method about the Cladosporium from the pine tortoise scale. The related reference is absence. There is no scale bar in the figure.
Author Response
Thank you for your positive judgment. As for the specific comments, we observe that:
1) The available information about possible mutualistic interactions is examined in section 4. Of course, a whole chapter was dedicated to entomopathogenicity due to the higher amount of data which have been published on this aspect.
2) We could not add a column where to report the order by reason of insufficient space. Therefore, we decided to use different colors to discriminate among orders, and to list the insect species in alphabetical order, which makes it easier for the reader to check if there are previous reports concerning a given species.
3) Figure 2 was added just to show how a Cladosporium colony looks like in the course of isolation from an insect. Reference to this figure is provided at page 9. No scale bar is necessary, since information about the colony or the insect size has no relevance.
Reviewer 2 Report
Comments and Suggestions for Authors
Dear authors,
The review “Cladosporium – Insect Relationships” is a well-written text and presents relevant information about the relationships between species of fungi of the genus Cladosporium and insects. The review is based on updated literature and also classic articles on Cladosporium, presenting relevant aspects of taxonomy, occurrence, hosts, entomopathogenic potential, and interactions with biological control agents and plants. I believe in the relevance of this manuscript, as it compiles 246 references on the topic, providing the reader with important information about the genus Cladosporium.
Congratulations on the manuscript.
Author Response
Thank you very much for your positive judgment and considerations about the relevance of our paper.
Reviewer 3 Report
Comments and Suggestions for Authors
This is a very well-written review of the interaction between Cladosporium and insects, arising from a careful examination and evaluation of the literature. It yields significant results, suggesting that the role of Cladosporium as an entomopathogen is likely much more important than previously thought. The authors dedicate ample space to this aspect, and ultimately, the conclusion supports this notion. Since the highlighted role of Cladosporium as an insect antagonist is emphasized, I would suggest that the authors consider highlighting this in the title as well, by adding the potential of Cladosporium as an antagonist or pathogen.
Author Response
Thank you very much for your positive judgment and considerations about the relevance of our paper. However, we prefer not to change the title to avoid emphasizing the entomopathogenic aptitude, also considering comments by reviewer #1.